# Current Technologies to Accelerate the Aging Process of Alcoholic Beverages: A Review

**Roselini Trapp Krüger** , **Aline Alberti** and **Alessandro Nogueira** *

Graduate Program in Food Science and Technology, State University of Ponta Grossa (UEPG),
Av. Carlos Cavalcanti 4748, Ponta Grossa CEP 84030-900, PR, Brazil
* Correspondence: nogueira@uepg.br; Tel.: +55-42-3220-3775

**Abstract:** The aging process contributes to the sensory evolution of alcoholic beverages, producing changes in the color and flavor of the final product. Traditionally, aging has occurred by storing beverages in wooden barrels for several months or years. To meet the demand for aged beverages, there is a need for large storage areas, a large number of wooden barrels, and, consequently, large volumes of stored product. Evaporation losses can also occur. In addition to the reactions of the beverage itself, there is also a transfer of wood compounds to the drink, which is later modified by successive oxidation reactions. This study addresses the alternative methods for accelerating the aging stage of beverages. These include the use of wood fragments, ultrasound, micro-oxygenation, pulsed electric field, high hydrostatic pressure, and microwave and gamma irradiation. These methods can be applied to optimize the process of extracting wood compounds, promote free radical formation, reduce oxidation reaction time, and accelerate yeast autolysis time. This study provides examples of some of the aforementioned methods. These technologies add value to the aging process, since they contribute to the reduction of production costs and, consequently, can increase commercial competitiveness.

**Keywords:** ultrasound; micro-oxygenation; pulsed electric field; wood fragments; high pressure; gamma irradiation



## 1. Introduction

Packing beverages in barrels was originally only used as a means of storage and transportation. It was possibly the Celts, around the third century BCE, who first used barrels to replace amphoras (ceramic vases), which tended to break very easily. The ancient Mesopotamians used palm wood to make barrels [1]). The use of wooden barrels dates back to the Roman empire, when they were used to store and transport wine, as well as other beverages, such as beer and oil, over long distances and for long periods [2].

Some alcoholic beverages (wines, spirits, beers, and liqueurs) can be stored in wooden barrels before consumption; during this period, the beverages develop sensory qualities that are appreciated by the consumers of the final product. These changes in the composition and concentration of chemical compounds in alcoholic beverages are caused by processes that take place between the wood of the barrel and liquid that is in contact with the wood during the aging period. Wood is a natural material that contains substances and aromas that are transferred from the wood to the liquid, giving the final product the desired organoleptic characteristics. Oxygen passes slowly through the pores of the wood, favoring the oxidation of certain compounds. At the same time, volatile components in the highest concentrations evaporate through the wood, while other substances can precipitate to the bottom of the barrel. Some of the main substances are furanic derivatives, lactones, volatile phenols, phenylketones, and phenolic aldehydes [3].

The production of high quality, aged, distilled beverages with guaranteed authenticity requires knowledge of the chemical compositions that develop during maturation, which are known as aging markers, based on the non-volatile profile. Statistical models suggest

that this transfer of compounds from wooden barrels is not constant and homogeneous throughout the period of maturation [4].

In the case of wine, successive physical, chemical, and biological transformations occur, thus improving the wine's stability and sensory characteristics. The aging that occurs in wooden barrels contributes to an increase in the complexity of the aroma, as well as changes in color and a reduction in astringency [5]. During this step, the volatile compounds extracted from the wood have a distinct impact on the aroma of the wine. The extraction of the volatile compounds from oak barrels depends on the quantity of compounds that are potentially extractable, as well as the contact time between the wine and wood [6].

Aspects such as the wood species, toasting intensity, container capacity, and maturation period are factors that can modify the chemical composition and sensory properties of alcoholic beverages [7]. The compounds that can be extracted from wood are finite; the extraction rate and amounts of compounds extracted decrease as the barrel is used repeatedly [8].

Technologies have been developed to accelerate the aging process, in order to provide more economical alternatives for alcoholic beverage producers, in addition to reducing the process costs, by minimizing the replacement or maintenance of the barrels [9]. There have been many scientific studies regarding alternative technologies for accelerating the aging stage.

Thus, this review discusses the emerging technologies used to accelerate the aging stage, such as the use of wood fragments, ultrasound, micro-oxygenation, pulsed electric field, high hydrostatic pressure, and microwave and gamma irradiation. The main forms of application are assessed, and some aspects are compared.

## 2. Research Methodology

To prepare this article, intense bibliographic research was carried out in specialized journals in the areas of food science and technology, as well as alcoholic beverages. The research area was existing processes and new technologies used to accelerate the aging of beverages in wood. The main scientific databases used were: ACS (American Chemical Society), Scopus, Web of Science, Science Direct, MDPI, Springer Link, and the Wiley Online Library. The bibliographic survey was carried out between 2010–2022, and 72 articles were found.

## 3. Alternative Technologies to Accelerate the Aging of Beverages

Table 1 shows a comparative summary of the advantages and disadvantages of each technology, the forms of application, and the main effects of each of them.



**Table 1.** Comparison of different aging technologies.

| Technology | Advantages | Disadvantages | Applications | Main effects | Reference |
|---|---|---|---|---|---|
| Wood fragments | Low cost—wood is reused. Testing new botanical varieties with low investment. Increase very old barrels. Possibility of dosage variation. | De-standardization of formats, and lack of knowledge, regarding the sensory contribution. Need for combined use with MOX. | Inside wooden barrels and steel barrels. | Migration of compounds from wood to beverage. | [10,11] |
| Ultrasound | Decrease aging time. Pre-treatment of wood fragments. Easy of scale-up. Processing old barrels. | Specialized equipment required. Ultrasound can produce rapid isomerization of compounds and oxidation reactions when applied intensively, thereby compromising sensory aspects. | On wood fragments. In bottles with product. In the beverage. Used in lees. | Reduction in activation energy of chemical reactions. Reduced contamination. Change in electrical conductivity. Induction of free radical formation. | [12–14] |
| Micro-oxygenation | Simulates gas transfer in barrels. Possibility of using steel tanks for aging and not just wooden barrels. Combined use with wood fragments. | Specialized equipment required. When unregulated, it can oxidize the drink in excess, sensory compromise | In beverages inside steel tanks, and in combination with wood fragments. | Presence of $O_2$ for yeast during fermentation. Reduction in addition of $SO_2$. Improved color. Reduced contamination. | [15] |
| Pulsed electric field | Decreased aging time. Use of grapes in pre-fermentation. The use in wines, in addition to accelerating esterification, also acts on contamination. | Specialized equipment required. It can give the product a metallic flavor. | Used in beverage. Used in barrel with beverage. Used on grapes. Used in lees. | Reduction in energy activation of chemical reactions. Reduction in addition of $SO_2$. Increased extraction of compounds from grapes. | [16–18] |
| High hidrostatic pressure | Decreased aging time. Low cost equipment. No heating. | If time is prolonged, it can destroy aromatic compounds. | In the young beverage before aging. Use on fruits. | Reduction in energy activation of chemical reactions. Increased extraction of compounds from grapes. | [19–21] |
| Microwave | Decreased aging time. Microbiological control. | Specialized equipment required. Hot spots. Difficult to scale-up. | Used on grapes. Used in beverage. Used on barrels. | Increased extraction of compounds from grapes. Induction of free radical formation. Reduced contamination. | [17,22] |
| Gamma irradiation | Decreased aging time. No heating. Contactless physical method. | Specialized equipment required. Consumers may fear the presence of radioactive waste. | Used on grapes. Used in beverages. Used in fragments and wood barrel. | Increased extraction of compounds from grapes. Induction of free radical formation. | [23,24] |

### 3.1. Wood Fragments

In order to take advantage of the residual material from the manufacturing of oak barrels or old discarded barrels, producers started testing the use of wood fragments, adding them directly to the wine or distillate in stainless steel tanks. In addition to making the business environmentally sustainable, it was found that this practice accelerated the aging process [25]. In 2006, the European Union authorized the use of oak wood chips in wine production, by Regulation (EC) N_o. 1507/2006 and Regulation (EC) N_o. 2165/2005. From that moment on, the addition of oak chips was intensified to obtain wines with characteristics similar to wines aged in wooden barrels [7].

The use of wood chips for aging reduces the redox potential, causing an increase in the concentration of esters and alcohols in a shorter time; a faster extraction of volatile compounds gives rise to wines with more intense colors and aromatic notes that are different from those achieved by the traditional method. This can be explained by the greater surface contact between the liquid and wood [3].

A wide variety of wooden pieces from different botanical origins, with various sizes and different degrees of burning, can be purchased commercially, in the form of chips, cubes, powder, shavings, granules, blocks, or boards. There are a variety of terms used to designate these different forms of wood, as well as a lack of standardization of these products. Knowledge of the chemical composition of these wood fragments (with and without toasting processes), mainly their antioxidant activities and ellagitannin content, is important for determining the effects on the beverage [10].

In addition, the prolonged and successive use of barrels contributes to reducing the amount of extractable substances, as well as the rate of extraction of these compounds [26]. Garde-Cerdán and Ancín-Azpilicueta (2006) conducted a review of experimental studies of wine aging in new barrels, with three and five reuses. They found a reduction in the availability of volatile compounds derived from the wood. The porosity and permeability were also altered [27]. The wood absorbs wine, due to the impregnation of the barrel walls, which causes progressive clogging of the pores, thus decreasing the rate of oxygen consumption by the wine [8].

Studies using wood fragments have made it possible to evaluate unusual wood species, different levels of toasting, and different dosages, as well as testing the different shapes of wood fragments, in relation to red wine and spirits, in order to verify the different phenolic and sensory contributions that can be obtained.

Rodríguez Madrera et al. (2010) evaluated different wood species that are not usually used in the manufacture of spirits for four weeks. Extracts obtained from maceration with chestnut, Spanish oak, and cherry chips showed phenolic and furanic profiles, total phenols, and antioxidant activities equivalent to commercial oak (American and French) [11].

Laqui-Estaña et al. (2019) evaluated the influence of time, wine variety (Carménère and Cabernet Sauvignon), and wood shape (chips, staves and barrels) on the phenolic composition of wine during aging in oak wood for one year. The extraction of phenolics was greater with chips, compared to barrels. In terms of sensory analysis, sweetness was less present in the wine aged with chips, and tannins were more prominent. In wine with aged with wood chips, the most influential phenolic compounds were vanillin, p-hydroxybenzoic acid, epicatechin, p-coumaric acid, and ferulic acid. For wine aged in barrels, the most important compounds were caffeine and syringic acids. Overall, the three factors were significant, in relation to aging [28].

Espitia-López et al. (2015) also carried out an experiment in red wine. They compared the use of wood fragments with traditional aging in wooden barrels, but with only one type of wine (Merlot) and between one type of fragment and wooden barrel, for six months. In the wines aged with chips, the phenolic compounds vanillin, p-hydroxybenzoic acid, epicatechin, p-coumaric acid, and ferulic acid were more present. For the wines aged in barrels, the most important compounds were caffeic and syringic acids. The extraction of phenolic compounds was higher and quicker in the wine aged with chips, compared to barrels. At six months, the sweetness was less present in the wine aged in chips [29].

Rodríguez-Solana et al. (2017) performed experiments with French and American oak, with four levels of toasting, three dosages (5, 15, and 25 g/L), and two types of format (chips and granules) of fragments in spirit wine. The best results for the concentration of phenolic compounds, color parameters, and antioxidant capacity were obtained with the wood in granular format and a medium level of toasting [30].

Dumitriu et al. (2019) also carried out experiments in red wine with French and American oak, but only varying the dosage (3 and 5 g/L). The samples that received French oak chips showed higher concentrations of furfural, 5-methyl furfural, guaiacol, 4-vinyl guaiacol, and trans-whiskey lactone, up to a dosage of 3 g/L. The samples using American oak chips contained higher concentrations of cis-whisky lactone in the wines treated with both dosages [7].

Woods other than traditional oak have also been studied, using wood fragments to identify the aging markers. Alañón et al. (2018) studied the volatile and non-volatile compounds in young white wine subjected to maturation with wood chips for four weeks, as well as wood barrels for four months with acacia wood, followed by sensory evaluation. The authors cited the complexity of identifying the markers; they found similarities with oak wood, but in less quantity, and noted how fundamental it is for sensory analysis to be carried out in the experiments, in order to contrast with analytical data [5].

In general, studies performed with wood fragments seek to evaluate the impact on color, content of total and individual phenolic compounds, antioxidant capacity, and interference in the sensory complexity of the beverage over short and long periods of time. Factors such as the type of fragment used, dosage, intensity of toasting (compared to traditional treatments), and use of wooden barrels are normally considered. The types of grapes used in wine experiments are also different in some cases, which does not help when comparing studies.

### 3.2. Physical Methods

Unlike wood fragments, which are a variation of wood barrels, physical methods require specialized devices that act by altering the conditions of the environment, as well as interfering with chemical reactions, and, consequently, with the aging process.

### 3.2.1. Ultrasound

The application of ultrasound to wine can promote the interaction of its ingredients, generating chemical reactions and accelerating reaction rates, which can promote chemical and structural changes equivalent to those that occur in traditional aging over several years [31]. The use of ultrasound as a technique to accelerate the aging of alcoholic beverages, such as wine and spirits, has been tested in recent studies (Table 2).

**Table 2.** Research regarding the application of alternative technologies in the aging of beverages.

| Technology | Alcoholic Beverage | Experiment/Parameters | Main Results | References |
|---|---|---|---|---|
| Ultrasound | Red wine (Tempranillo) and model wine | US application in the beverage. Ultrasonic homogenizer. 24 kHz, 400 W, 50 μm, 52 W/cm$^2$. Application time: 170 min. 1st stage: 5 min—2 times/week—5 weeks. 2nd stage: 15 min—2 times/week. Aging time: 135 days. | The application of US on lees doubled the increase in the polysaccharide content. Volatile fermentative compounds and total anthocyanin content were affected. The sonification of lees before dosing in wines may be a more suitable procedure. | [32] |
| | Red wine (Cabernet Sauvignon) | US application in the wine before storage in glass bottles. Ultrasonic homogenizer. 20 kHz, 00/150/200 W. US application time: 0/14/28 min Storage time: 70 days. | Ultrasonic treatment not only temporarily influenced the color characteristics and phenolic compounds of the wine, but also had a longer effect on its evolution during wine storage. Treated wine showed faster changes than untreated wine, concerning the studied parameters. | [33] |
| | Coffee liqueur | US application in the beverage. Ultrasonic bath. 20 kHz. 0 to 6 h. | After six hours of US application, the alcohol content, caffeine content, and turbidity were similar to conventional aging after 180 days. | [34] |
| | Greengage wine | US application in the beverage. Ultrasonic bath. 28 kHz and 45 kHz. 240/300/360 W, 50 min. Storage time: 15 days. | The best conditions for applying the US (360 W, 45 kHz for 30 min) resulted in an increase in the concentration of acids and esters, as well as a decrease in secondary alcohols. | [35] |
| Micro-oxygenation | Red wine (Cabernet Sauvignon) | Application before barrel aging. Dosage: 3 mg/L per month. Time: 3 months. | Produced wines with less intense red color. Increased anthocyanin concentration and reduced astringency. | [36] |
| | Red wine (Cabernet Sauvignon) | Application before and after malolactic fermentation (two stages). 1st stage: 20 days: 15 mg/L. 2nd stage: 3 months: 6 mg/L. Time aging: 20 months in barrels and bottles. | There was a significant acceleration in the kinetics of anthocyanin degradation and transformation reactions, with little or no impact on the normal evolution of its tonality. These effects were maintained over time, so that the differences between the micro-oxygenated and control wines remained after 20 months of aging. | [37] |

**Table 2.** *Cont.*

| Technology | Alcoholic Beverage | Experiment/Parameters | Main Results | References |
|---|---|---|---|---|
| | Red wine | Before bottling. 3 mL/L/month. Time: 3 months. | After three months, wines with MOX showed a chromatic and phenolic profile similar to wines aged in oak barrels. However, after six months of bottling, wines with MOX did not show the same chromatic similarity, with an increase in yellow color. | [38] |
| | Red wine (Cabernet Sauvignon) | Applied in wines with four different microbiological compositions. Before bottling in transparent glass bottles. 1.04, 2.35, and 3.65 mg/L per year. Time: 12 months. | The levels of acetaldehyde in bottling affected the phenolic profile after one year. Increased levels of acetaldehyde and oxygen intake also produced higher levels of heterocyclic acetals of glycerol. | [39] |
| Pulsed electric field | White wine (Chardonnay) | Application on yeasts. 5 and 10 kV/cm. Pulse: 75 µs. Time: 6 months. | There was no significant difference in the total phenol content, total volatile acidity, pH, ethanol, wine color parameters, reduction of turbidity, foam formation, and interaction with tannins between wines that received yeasts treated, or not treated, with PEF. The wine that received yeast treated with PEF released mannoproteins in one month were equivalent to wine with untreated yeasts in six months. | [40] |
| | Red wine (Merlot) White wine (Chardonnay) | Bottle application. 6, 12, 18, and 24 kV/cm. Pulses: 0, 100, 200, and 300 µs. Time: 210 days. | Lower levels of PEF may have been more effective than high levels of PEF in changing the concentration of organic acids that occurred during bottle storage. The strength of PEF applied to white wine during bottle aging was lower than that of red wine. | [41] |
| | Spirit wine | On wooden barrels (2 and 5 L) with beverage. 1 kV/cm–50 Hz. Time: 12 h. | The content of tannins, total phenols, and volatile phenols was significantly increased after treatment with PEF, in both tested barrel sizes, and was also higher than 225 L barrel-aged beverage after 14 months. | [42] |
| | Red wine (Cabernet Sauvignon) | Application on grapes. 50 kV/cm. 50 pulses–3 µs. Time: 14 months. | The best chromatic characteristics and highest phenolic content were obtained by treatment with PEF. This effect remained after the fermentation process or even increased during aging in oxidative conditions in American oak barrels, as well as subsequent storage in bottles. No sensory differences in color and bouquet were detected after eight months of aging in bottles. | [18] |

**Table 2.** *Cont.*

| Technology | Alcoholic Beverage | Experiment/Parameters | Main Results | References |
|---|---|---|---|---|
| High hydrostatic pressure | Mei liqueur | Application on Mei fruit. 600 Mpa. Time: 5 min. | Mei liqueur showed significantly higher ΔE, sugar, and alcohol concentrations during 180-day maceration. Electron microscopy revealed that treatment with HHP damaged the cellular structure of Mei fruits. | [43] |
| | Red and white wine | Application in young wine. 200, 400, and 600 MPa. Time: 5, 15, and 25 min. | Higher pressures resulted in greater perception of changes. The best sensory results were in white wines after 12 months of aging. The immediate effects after treatment were a reduction in individual phenolic compounds and increase of color parameters in red wines. Treatment with HHP helped to reduce the use of $SO_2$ in wine production. | [21] |
| | Sorghum spirit | 50 MPa. 5 cycles. 1 h. | An increase in ethyl acetate, reduction in methanol, and increase in fruity, floral, and sweet aromas in the beverage, as well as greater sensory acceptance of the distillate at the end of the process. High pressure improved and accelerated the activation energy of the esterification reactions. | [44] |
| | Red wine | Application in young wine. 500 MPa for 5 min. 600 MPa for 20 min. Storage time: 5 months. | High-pressure treatments influenced the phenolic composition of red wine, mainly altering the aroma. Most of these effects were noticed after five months of storage; the most pronounced effects were for the pressure treatment of 600 MPa for 20 min. A lower content of monomeric anthocyanins, phenolic acids, and flavonols in pressurized red wine after five months of storage was due to the increase in the condensation and oxidation reactions of these compounds. Polymerization and cleavage reactions of proanthocyanidins also occurred. | [19] |
| Microwave | Red wine and model wine | Time: 3, 5, 10, 15, and 20 min. Temperature: 30, 40, 50, 60, and 70 °C. Power: 100, 200, 300, 400, and 500 W. | The intensity of free radical release was higher with greater potency and longer time; however, there was a decrease with increased temperature. Recommendation to use lower temperature and power for 10 min. | [45] |

**Table 2.** *Cont.*

| Technology | Alcoholic Beverage | Experiment/Parameters | Main Results | References |
|---|---|---|---|---|
| | Red wine (Merlot) | Application on grapes. Fruit ripening levels: 21.16, 23.14, and 25.16° Brix. Power: 1200 W. Time: 10 min. Temperature: 40 °C. | MW treatment produced positive improvements in anthocyanins, tannins, total phenolics, polymeric pigments, and color, when applied to unripe Merlot fruit with 21.1 Brix. There was a lesser effect on wines made from ripe (23.1 Brix) and fully ripe (25.1 Brix) fruit. In the case of fully ripe fruit, the application of MW caused negative effects in some phenolic compounds. The positive results, observed when MW was applied to unripe fruit, suggests that this technique furthers phenolic extraction, in the case of fruit that may be deficient in phenolic maturity. | [46] |
| | Red wine (Pinot Noir) | Application on grapes. Household microwave. 1150.W (four times). Water bath: 70 °C/1 h. | Alcoholic and malolactic fermentation were complete at 17 days post-inoculation for all three yeast treatments. At 16-months bottle age, the AWRI1176-treated wines had approximately twice the non-bleachable pigment and color density of wines fermented by EC1118 and RC212. | [47] |
| | Red wine (Cabernet Sauvignon) | Time: 5, 10, 15, and 20 min. Temperature: 40, 50, 60, and 70 °C. Power: 100, 300, 500, 700, and 900 W. | The power and time of irradiation were the factors that had the greatest influence. The use of lower temperature and longer time are recommended. The suggested conditions for the application of MW in red wine processing are 500 W power, temperature equal to 40 °C, and an exposure time of 20 min. | [48] |
| Gamma irradiation | Distilled baijiu | Doses: 10 (600–7600 Gy). | It took 28 days, with an optimal dosage of 5.9 kGy, for the system to reach a stable state, compared to natural aging, with the addition of aromatic organic compounds and decline of other undesirable organic compounds, due to the formation of a large amount of active free radicals. | [49] |

**Table 2.** *Cont.*

| Technology | Alcoholic Beverage | Experiment/Parameters | Main Results | References |
|---|---|---|---|---|
| | Cachaça | Application:<br>Cachaça, barrel, and cachaça + barrel.<br>Six experiments.<br>150 Gy (50 Gy/min).<br>Time: 390 days. | The sensory evaluation found that the irradiated cachaça and/or barrel received the highest approval rating (aroma, flavor, and appearance) and that there was an acceleration of the aging process. The concentration of tannins and aldehydes were higher in the treatments, both in the cachaça and irradiated barrel; in these treatments, the highest color intensity was verified. | [50] |
| | Rice wine | Doses: 200, 400, 600, and 800 Gy (20 Gy/min).<br>7 days (25 °C). | There was an increase in ethyl acetate and decrease in polyols. There was an improvment in some rice wine defects, and the production of a higher taste quality in the rice wine, without the presence of irradiation residues. | [24] |
| | Red wine (Merlot and Traminer) | Application on grapes<br>Doses: 670, 1300, 2000, and 2700 Gy (6,4 Gy/min)<br>18 °C | There was a negative impact on the content of amino acids in the musts. However, wines produced from irradiated grapes showed no loss of quality in basic chemical composition. Volatile acidity was lower at higher irradiation doses, compared to control wines. The irradiation dose of 2700 Gy showed the best results, regarding anthocyanin concentrations in Merlot wines. Other phenols, such as flavonols and flavonols, were not affected by irradiation. | [23] |

Note: US = ultrasound; MOX = micro-oxygenation; PEF = pulsed electric field; HHP= high hydrostatic pressure; MW = microwave.

The effects of ultrasound in liquid systems are mainly related to the phenomenon of cavitation, which occurs when longitudinal waves are created, forming alternating regions of compression and rarefaction in the molecules. Ultrasound systems are classified, according to the form of application, as either low intensity ($<1$ W/cm$^2$), which employs high frequency ($>1$ MHz), or very low and high-intensity power levels (10–1000 W/cm$^2$), using low frequencies (20–100 kHz) and high power. Low-intensity systems produce minimal physicochemical changes in the material through which the wave passes; in contrast, high-intensity systems generate cavitation and produce physicochemical modifications in numerous applications [51].

The ease of reproducing laboratory results on an industrial scale is a positive aspect of ultrasound by making adjustments to intensify the process and reduce energy consumption. The choice of the system will depend on the matrix and what purpose is desired. In extractive processes, when assisted by ultrasound, cavitation breaks the membranes of plant cell walls and other structures, thus facilitating the release of extractable organic compounds, consequently reducing the time employed and solvent consumption, simplifying the processes, giving greater purity to the final product, eliminating wastewater post-treatment, and reducing fossil energy consumption [12].

Another relevant aspect of the use of ultrasound in liquid systems is the formation of free radicals, through a phenomenon called sonolysis. During this process, pure water is saturated with an inert gas and undergoes thermal homolysis of the water molecules inside the bubbles, thus producing the radicals H and OH, which can recombine to form $H_2O$ again, as well as $H_2$ and $H_2O_2$ [52]. The presence of free radicals produced by ultrasound can induce oxidative reactions, which contribute to the complexation of flavors and aromas of aged drinks, with the presence of phenolic compounds and volatile compounds, from the reaction between the young beverage and wooden barrels [14].

Yan et al. (2017) evaluated the levels of the ions and ionic strength to investigate the mechanism of change in the electrical conductivity of wine under ultrasonic irradiation. Studies regarding changing electrical conductivity during the aging process in wines have represented a scientific approach to quantify the chemical reactions related to aging because these reactions can be initiated by the conversion of cations and degradation or auxiliary function of organic acids. The change in electrical conductivity is related to the high temperature localized from the collapse of bubbles, due to the effect of ultrasonic cavitation or movement of ions accelerated by ultrasonography. These changes in ions, especially for the cations of the transitional elements and anions of some organic acids, are important because they can initiate the redox reactions of the phenols that occur in the natural aging of wine [53].

The intensity of ultrasound treatment may also have non-positive aspects. Jambrak et al. (2017) investigated the effect of ultrasound power (50%, 75%, and 100% amplitude) and temperature increase on antioxidant capacity, color, total phenolic compounds, degree of non-enzymatic browning, and sensory properties by using electronic tongue, in relation to blueberry nectar. Treatment at 50% amplitude for three and six min (softer) was considered the best result. The sensory evaluation resulted in the conclusion that ultrasound-treated nectars received lower scores (for taste, odor, aroma, and color), compared to untreated samples. Ultrasound can produce the rapid isomerization of the compounds and oxidation reactions (which occur as a result of interaction with free radicals) that are generated during ultrasonic treatment. The results also showed that 100% amplitude had a greater impact on non-enzymatic browning, which can be explained by a higher concentration of anthocyanins, which are degraded as a result of Maillard reactions [13].

### 3.2.2. Micro-Oxygenation

Micro-oxygenation (MOX) is a technology where minimal amounts of oxygen, at rates of mg $O_2$/L, are intentionally dissolved in 'young' beverages, in an attempt to simulate the oxygen transfer model that occurs through the porosity of wooden barrels, in order to cause desirable changes in color, aroma, and texture. In winemaking, the main effects

of micro-oxygenation include an improvement in the vitality of yeasts during alcoholic fermentation, improvement in the color and stability of the wine, more complex sensory characteristics, reduction in sulfur odors, and ability to simulate the reactions that occur during wine aging [15].

The use of micro-oxygenation involves treating a wine with well-controlled sub-saturation doses of oxygen over short periods of time. A very large oxygen flow may be inadequate, thus leading to the oxidation of aromas, precipitation of high molecular weight polymers, and reduction in intensity of the color of the wine. The proper use of micro-oxygenation can stabilize the color and decrease the astringency and herbaceous characteristics of wine [38]. When micro-oxygenation is applied, oxygen doses must be individually adapted to each wood product, due to particularities, such as botanical origin and degree of toasting, thereby avoiding excessive amounts that can reduce aromatic complexity and result in undesirable oxidation processes [54].

The presence of oxygen is fundamental in the formation and development of color during the aging of wine spirits, as well as the influence of the MOX level, which promotes a more intense extraction of phenolic compounds and furan aldehydes. Canas et al. (2022) tested a more intense initial level of MOX at 2 mL/L/month, varying the application period for 15, 30, and 60 days, and then maintaining the flow rate for 0.6 mL/L/month, until 365 days were completed. The results showed that the stronger oxygenation for 60 days promoted significantly faster color evolution and higher TPI, probably by favoring the leaching of the outer layers of the wood and extraction/degradation of tannins [55].

In similar study, Oliveira-Alves et al. (2022) investigated the influence of bottle storage on the antioxidant activities and phenolic composition of wine spirits aged with chestnuts staves and micro-oxygenation. After 12 months, the MOX modality, with greater oxygen application, showed better performance, thus reinforcing the idea that this technological option seems to be the most suitable for the quality of the wine spirit and sustainability of aging [56].

Most studies focus on wines, but Canas et al. (2019) carried out comparative experiments on the effects on phenolic composition and color acquired by wine spirits using the traditional method and an alternative method with MOX and boards for six months. The alternative method showed a faster extraction of wood-derived compounds in the wine spirit, with a higher amount of phenolic compounds of low molecular weight and total phenolic content than the wooden barrels. The alternative method also resulted in greater evolution of chromatic characteristics (lower luminosity, greater saturation, and higher intensities of red, yellow, and brown tones) [57]. Table 2 shows other examples of research performed using MOX technology in alcoholic beverages.

### 3.2.3. Pulsed Electric Field

Pulsed electric field (PEF) technology consists of the application of an electric field, in the form of high voltage pulses. The voltage is conducted between two electrodes for a short period, usually microseconds. This PEF treatment, also known as electroporation or electro permeabilization, is considered a non-thermal process, and it reduces the risk of the thermal degradation of thermolabile compounds [16,22].

The first studies regarding the use of pulsed electric fields referred to its use to reduce the microbial load in foods, which occurs due to the damage caused to the integrity of the cell membrane of the microorganisms that are present, which is induced by high voltage pulses [58].

Pulsed electric fields have been used in winemaking to intensify the extraction process of phenolic compounds from grape skins, which occurs during the maceration stage, with prior application on fruits. This treatment intensifies the chromatic characteristics, increases the content of volatile compounds, and improves the sensory attributes, which are perceived in wine after aging in wooden bottles or barrels [16,18]. Table 2 shows the application of PEF in alcoholic beverages and main effects, which have been developed in some scientific studies.

In addition, the application of PEF has been studied as a technique to replace the addition of sulfur dioxide ($SO_2$), which is usually used to reduce the risk of microbial

contamination in wines. The antimicrobial effect of PEF, due to the electroporation of microorganism cell membranes can selectively prevent contamination from wild yeasts and bacteria [17].

Another relevant aspect provided by the use of PEF is the intensification of the esterification reactions between ethanol and acetic acid. When an external electric field is applied, the molecules absorb this energy, providing an additional vibration of hydrogen bonds, decreasing the activation energy, and, consequently, favoring and accelerating the rupture of the chemical bond [59].

During the stage of wine aging, the presence of mannoproteins, generated from the autolysis of the strains of *Saccharomyces cerevisiae*, has been studied and related to increased stability in wines. Their presence contributes to reducing the formation of turbidity, prevents the precipitation of tartar salts, reduces astringency, increases flavor and aromas, and stabilizes color. In a traditional wine aging process, with the presence of lees, the release of mannoprotein molecules occurs over long periods. The application of a pulsed electric field has been studied as an alternative for accelerating this step by accelerating the autolysis of yeast strains, due to the generation of the electroporation of their cell membranes [40].

A recurring negative aspect in studies, regarding the application of PEF in foods, is the migration of metal ions from the electrodes used, thus providing a metallic flavor to beverages and foods, with an increase mainly in the concentration of iron ions [16].

### 3.2.4. High Hydrostatic Pressure (HHP)

High hydrostatic pressure (HHP) is a technique widely used to preserve and modify food products. HHP treatment is non-thermal and can be applied to foods, either with or without packaging, at high pressure in the range of 100 to 600 MPa. The main purpose of HHP, in this context, is to inactivate undesirable microorganisms and enzymes; due to the minimal increase in temperature, there is little effect on the low molecular weight compounds during processing [21]. HHP can be considered a green technology because it uses water as a compression medium, and it is energy efficient [60].

When a liquid product is forced through a narrow space in a few seconds, and then subjected to rapid acceleration, the product is subjected to strong shear stress, cavitation, and turbulence in the medium. This type of device is called a high-pressure homogenizer. When this type of physical energy is applied in liquids, it can increase the collision frequency and can contribute to the esterification of spirits, thus accelerating the maturation of this type of beverage [44].

In beer processing, high pressure can be used in the stages of malting, maceration, boiling, filtration, and pasteurization, which can generate effects on physicochemical, microbiological, and sensory characteristics. Beer processed under high pressure can have a prolonged shelf life, due to the inactivation of microorganisms that are harmful to beer. This technique can modify protein structures, such as the enzymes that are present in malt, such as α- and β-amylases, and promote saccharification, as well as being able to isomerize the hops and increase the bitterness of beer. Using HHP in the process can result in a reduction in the demand for steam, as well as the generation of waste [61].

Santos et al. (2019) carried out five experiments with red wine; they compared the aging processes using the following: conventional treatment (barrels); wood chips; micro-oxygenation + wood chips; pressurization; and control. The aforementioned study found a similar degree of polymerization of tannins after five months of bottling in wines that received pressurization, compared to traditional systems. The authors noted that high pressure promoted greater oxygen diffusion in wine; when it is used in combination with chips and MOX, it can present promising results for the production of red wines [20].

The use of HHP technology for modifying wine composition can benefit the wine industry, especially for improving wines with low aging potential. HHP can be potentially used as an enological practice, modulating the organoleptic properties of wine by decreasing astringency and increasing pleasant aromas [19]. Other studies are shown in Table 2.

### 3.2.5. Microwave

Microwave (MW) technology has been applied in the food industry to reduce processing time and contribute to food preservation. Microwaves are non-ionizing electromagnetic waves, with frequencies between 300 and 300 GHz; smaller bands are used for industrial purposes. Microwaves have a double effect on matrices: molecular movement through the migration of ions and the rotation of molecules, with the formation of dipoles. The resistance offered to the migration of ions and realignment of the dipoles generates friction forces that give rise to heating, without changing their structures, if the temperature is not too high [62].

The power dissipated in the medium depends on the dielectric properties and strength of the electric field. In conventional heating, heat transfers take place from the heating device to the medium, while, in microwave heating, heat is dissipated within the radiated medium, with a much faster rise in temperature. The maximum temperature of a microwave-heated material depends on the rate of heat loss and applied power. In transparent media, the occurrence of standing waves results in "hot spots" if energy dissipates faster than the heat transfers to cooler neighboring areas [22].

Microwaves with high frequency electromagnetic waves destroy the stability of weak hydrogen bonds, thus increasing the rotation of polar molecules. This induction of the dipole rotation of the molecules and migration of dissolved ions intensifies the production of free hydroxyl radicals [48].

Tests carried out using microwaves at medium intensity (700 W) and temperature control, in the maceration of crushed grapes on a laboratory scale, increased the amounts of varietal compounds in the must in a very evident way. After sensory analysis, the wines that were most valued by the tasters were those treated with MW because they presented a greater intensity of aroma and floral odor, especially those made without $SO_2$, which shows that treatment with MW can be very suitable for increasing the aromatic potential of wines [62]. Table 2 shows other studies that have used MW, with summaries of the main effects.

When microwaves are applied directly to wooden barrels during the aging stage, the pores of the wood are opened, which increases the supply of oxygen and favors the transfer of certain substances from the wood to the aged wine product. This results in an improvement in product quality, as well as a reduction in the time required to complete the aging stage [3].

A factor to be taken into account, regarding microwave technology, is the unforeseen in scale-up, which may compromise its advantages. The microwave power dissipates the greater the distance from the reactor (penetration depth), and this can result in significant surface overheating. For larger scale processes, continuous flow reactors are required. However, continuous flow scaling is associated with certain problems, such as non-uniform temperature distribution within the product, which is due to differences in dielectric and thermophysical properties, as well as non-uniform electromagnetic field distribution. In addition, controlling processing parameters, such as power, flow rate, temperature, and pressure, can also be critical [17].

### 3.2.6. Gamma Irradiation (GI)

Irradiation technology was approved as healthy, at doses of up to 10 kGy, for applications in food by the joint FAO/IAEA/WHO committee in the 1980s. The irradiation of food products is a physical treatment involving direct exposure to electron or electromagnetic rays produced by cobalt-60. It was first used against insects, delaying the ripening of fruit and destroying microorganisms, with action on DNA molecules. Irradiation can be carried out at ambient or lower temperatures, which ensures the better preservation of the nutritional values and physicochemical properties of foods [63].

The use of radiation in beverages has been studied as an alternative to minimize the loss of components, which occurs by thermal processes. Carvalho Mesquita et al. (2020) verified the efficacy and effect of gamma irradiation at different doses on the nutritional,

microbiological, and sensorial characteristics of grape juice blends during storage for 120 days. Grape juice blends (2.0 kGy) presented the highest antioxidant content and vitamin C increase until 90 days, compared to the other treatments. Sensory tests showed that the quality remained unchanged, as well as reduced fungi and yeasts, during storage at room temperature [64].

Gamma irradiation appears to be a potential method for achieving maturation quality, when used as an alternative method for maturing maize wine. Chang (2004) performed tests in maize wine by applying ultrasound and gamma irradiation during the aging stage. The ultrasonic wave treatments (20 kHz and 1.6 MHz) did not appear to be suitable methods for achieving the maturation quality requirement. The quality of maize wine that was gamma irradiated improved as the irradiated doses increased. The treatment with 800 Gy was rated nearly as good as one-year conventionally matured maize wine [65].

This technology has also been studied as an alternative to accelerating the aging of alcoholic beverages (Table 2); however, there is much less research regarding this technique, compared to the other technologies mentioned above.

## 4. Research Regarding the Combined Application of Alternative Technologies

Many scientific studies have examined the combined use of alternative technologies, with wood fragments and ultrasound or micro-oxygenation, as well as the presence of wood fragments in stainless steel tanks, at different processing stages. The joint application of these technologies has been shown to be more effective than when used in isolation [9,14,26,60,66–69]. Table 3 provides a summary of some of these studies.

The practice of aging wine in stainless steel tanks involves storing it in contact with the wood, with a small dosage of oxygen to obtain a final wine that is more stable over time, as well as with the same characteristics as wine aged in barrels. The dosage of dissolved oxygen is a determining factor for achieving correct wine development, and it must be applied according to the needs of the wine and type of wood chosen. Álamo et al. (2010) performed experiments to determine the oxygen demand for the same wine to be aged in tanks, from different sources of wood, in the shape of chips and staves made from American (*Quercus alba*), French (*Quercus petraea*), and Spanish (*Quercus pyrenaica*) oak. The results indicated that the wine treated with staves consumed more oxygen than the wine aged with chips. With regard to the botanical origin of the wood, the wine aged with French oak products required a higher dosage of oxygen [54].

Schwarz et al. (2014) performed an accelerated aging experiment at the laboratory scale of brandy from Jerez using oak chips and ultrasound as an extraction method. After 30 days, the method with twice the wood dosage, one hour of US application, and another 24 h of maceration produced brandy with analytical and sensory characteristics similar to brandy aged in the traditional manner for an average period of 6 to 18 months [70].

Nocera et al. (2020) compared the traditional method of aging in barrels with alternative technology, using wood chips with micro-oxygenation in stainless steel tanks. Chestnut and Limousin oak woods were tested for one year. Measurement of antioxidant activity, total phenolic index, and related phenolic composition of the wine spirit were performed. Significantly higher antioxidant activity (more 14% DPPH inhibition), together with greater enrichment in wood-derived compounds (more 24.07 total phenolic index), were achieved in wine spirits aged using the alternative technology, compared to the use of new barrels. Moreover, chestnut wood stood was better than Limousin oak wood, with higher levels of antioxidant activity and a higher phenolic index [71].

**Table 3.** Research regarding the application of combined technologies in the aging of beverages.

| Beverage | Combined Technologies | Parameters | Experiment | Main Results | References |
|---|---|---|---|---|---|
| Greek distillate | Ultrasound | Ultrasonic bath: 37 kHz-200 W, AED: 50 W/L. Total of 5 min and 72 h intervals. | Compared aging with a static model and US application. Time: 30 days. | No statistical differences were shown between the aging treatments. The antioxidant activity in the distillate treated with chestnut chips was highest. | [66] |
| | Wood fragments | White oak, acacia, cherry, and chestnut. Chips: $1 \times 1 \times 0.5$ cm. Toasting: untoasted. Dosage: 4.5 g/L. | | | |
| Plum distillate | Ultrasound | 400 W–24 kHz. 50% amplitude. Three minutes a day for five days a week. | Time: 12 months. System: static, with circulation and with ultrasound. Temperature: 18, 20, 35, and 45 °C. | Improved color and increased concentrations of phenolic compounds, which is important for aroma and flavor. These changes depended on the type and dose of the chips, as well as the maturation conditions. The changes were most intense with heating at 35 °C. | [26] |
| | Wood fragments | French oak and used barrel chips. Toasting: light. Size: $7.5 \times 10$ mm. Dosage: 3.5 and 7 g/L. | | | |
| Spirit wine (Holanda) 65 and 40% (ABV) | Ultrasound | US: 40 kHz and 40 W/L in periods (min): $6'/24'$ and $1'/4'$ (US/rest). | To evaluate the influence of US power, with and without pulse, and optimal aging time. Circulation of beverage: 40 and 50 L/h. Light: with and without. Temperature: 13 and 25 °C. Aeration: with and without. Time: seven days. | The 40 W/L power improved the extraction of phenolics by 33.9% after seven days of aging. The best aged distillate that was produced was obtained with the highest alcohol content, the largest amount of oak chips, at room temperature, and a high flow rate. The presence of oxygen in the samples and absence of light increased the quality of the beverage. | [14] |
| | Wood fragments | Oak. Dosage: 4 and 5 g/L. Toasting: medium. | | | |

**Table 3.** *Cont.*

| Beverage | Combined Technologies | Parameters | Experiment | Main Results | References |
|---|---|---|---|---|---|
| Spirit wine | MOX | Dosage O$_2$: 2 mL/L/ month. | Stainless steel tanks: 1000 L.<br>- With oak staves.<br>- With chestnut staves.<br>Time: 12 months in tanks + MOX.<br>Oak barrel: 250 L;<br>- Barrels from Limousin oak.<br>- Barrels from chestnut.<br>Time: 12 months. | The alternative aging system (chips + MOX) produced a greater number of volatile compounds and sensory descriptors than the traditional system. The most evolved aging characteristics in the attributes of color, aroma, and flavor were found in the samples aged using the alternative system, with improved extraction of volatile compounds from chestnut wood. | [67] |
| | Wood fragments | Limousin oak and chestnut. Shape: staves (91 × 5 × 1.8 cm). Dosage: 85 cm$^2$/L. Toasting: medium plus. | | | |
| Red wine (Tempranillo) | MOX | Dosage O$_2$. Two months: 2 mL/L/ month. | Tested different storage conditions. Stainless steel tanks, 250 L.<br><br>- With chips: with or without MOX.<br>- With staves: with or without MOX.<br>Time: six months in tanks + 18 months in bottles.<br>Oak barrel: 225 L. Time: 12 months in barrels + 12 months in bottles. | Chromatic parameters showed no difference between treatments. Volatile compounds (furfural and 5-methylfurfural) were superior in the first six months in the treatments with barrels and staves. Gallic and ellagic acids were higher in the treatment with chips. The optimum contact time between the fragments and wine can be estimated at two months. The best sensory quality of the wines treated with staves was obtained in short periods, while for those aged in barrels it was better with a longer time. | [68] |
| | Wood fragments | Chips and staves of American oak. Toasting: medium. Dosage: chips—4 g/L staves—0.4 m$^2$/hL. | | | |
| Cider brandy | MOX | Dosage O$_2$: 50 mL/L/ month. | Compared traditional barrel aging system with alternative aging. Time: 12 months. | The treatment with MOX accelerated the changes in cider distillates, when compared to traditional aging in barrels. A higher degree of oxidation in micro-oxygenated spirits favors the content of benzoic derivatives and total acetaldehyde. It additionally showed a higher degree of hydrolysis, resulting in a higher concentration of oak lactones and gallic acid and more pronounced decrease in the levels of 3-methyl-1-butyl acetate and 2-phenylethyl acetate. | [9] |
| | Wood fragments | French, American, and Spanish oak. Toasting: strong. Dosage: 178 cm$^2$/L. | | | |

**Table 3.** *Cont.*

| Beverage | Combined Technologies | Parameters | Experiment | Main Results | References |
|---|---|---|---|---|---|
| Red wine (blend: Cabernet Sauvignon +Merlot+Malbec) | MOX | Dosage $O_2$: 1 mg/L/ month. | Six experiments: 1- barrel, American oak. 2- barrel, French oak. 3- stainless steel tank: MOX. 4- stainless steel tank: chips + MOX. 5- stainless steel tank: staves + MOX. 6- stainless steel tank: tannin extract. (3 g/hL) + MOX. Time: six months (tanks and barrels) and another five months after bottling. | Differences in color were observed in the chemical composition of wines with MOX and added tannin. The effect was still evident after five months of aging in bottles. Although there were no significant sensory differences between treatments, in relation to taste, and the addition of oak affected the aromatic profiles of wines. MOX treatment with staves and wood chips, respectively, shared aroma attributes with French and American oak barrel treatments. | [69] |
| | Wood fragments | American and French oak. Format: chips, staves, and barrel. Toasting: medium plus and medium. Dosage: 1.63 g/L. | | | |
| Red and white wine | HHP | Total of 400 MPa. Five and 30 min. | Five experiments: 1- no treatment. 2- HHP—400 MPa—5 min. 3- HHP—400 MPa—30 min. 4- bottling after maceration, 45 days. 5- Maceration in tanks without wood fragments. | There was an increase in the content of polyphenols and an increase in the chromatic parameters in experiments 2 and 3, in relation to experiments 1 and 4, in white wines. In red wines, these effects were not observed. | [60] |
| | Wood fragments | Holm oak. Format: pieces 2–4 mm. Toasting: 165 °C/35 min. Dosage: 5 g/L. | | | |

Note: US: ultrasound; MOX: micro-oxygenation; HHP: high hydrostatic pressure; ABV: alcohol by volume; AED: acoustic energy density.

## 5. Final Considerations

Continuing research on alternative technologies for accelerating the aging stage of beverages is necessary, as there is still a lack of information, regarding the application of these new processes in different types of beverages, such as beers, spirits, liqueurs, and cachaças.

Further studies are required, regarding the sensory impact on beverages, especially in relation to the time gained in accelerating the aging stage and, consequently, financial impact of alternative processes, in relation to traditional processes.

The combined use of wood fragments and MOX has shown to be very promising; this technology has been more studied than some of the other technologies addressed in this article.

The constant search for technological innovation in the beverage industries, through the substitution of traditional models for more sustainable and more efficient processing practices, is necessary for maintaining competitiveness in the market.

**Author Contributions:** R.T.K. writing—original draft preparation, A.A. and A.N. writing—review and editing. All authors have read and agreed to the published version of the manuscript.

**Funding:** This work was supported by the National Council for Scientific and Technological Development (CNPq), under Grant #313417/2019-9, and Coordenação de Aperfeiçoamento de Pessoal de Nível Superior—Brasil (CAPES), under grant finance code 001.

**Data Availability Statement:** Not applicable.

**Conflicts of Interest:** The authors declare no conflict of interest.

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
