# Peer review of "Current Technologies to Accelerate the Aging Process of Alcoholic Beverages: A Review"

_beverages, doi:10.3390/beverages8040065_

Round 1

Reviewer 1 Report

The manuscript describes current technologies to accelerate the 2 aging process of alcoholic beverages. The manuscript is extensive and clearly written. The table shows the advantages and disadvantages of the new technologies. It also tabulates the main results of recent research, which makes this review scientifically significant. However, a similar review of the literature was recently published, which I would like to include in the manuscript (Soraya Solar, Remedios Castro & Enrique Durán Guerrero (2021): New Accelerating Techniques Applied to the Ageing of Oenological Products, Food Reviews International, DOI: 10.1080/87559129.2021.1934009).

I also suggest that more attention be given to describing the effects of new technologies on sensory properties, which may be questionable when using these physical methods. Unfortunately, very often in the manuscripts the results of the sensory analyzes are "hidden", precisely because of the impression of "cooked wine" that can result from the application of these technologies, while only the influence on phenols, color, etc. is indicated (see Jambrak, A.R., Innovative Food Science and Emerging Technologies (2017), http://dx.doi.org/10.1016/j.ifset.2017.04.015). Some of these shortcomings are listed in Table 1 (like metallic flavor), but perhaps this should be emphasized even more.

Author Response

Manuscript ID: beverages-1894695

Title: Current technologies to accelerate the aging process of alcoholic beverages: a review

Thank you for your response on our manuscript beverages-1894695.

The constructive criticism of the reviewer was much appreciated and we revised our manuscript accordingly. All suggestions were accepted, more information and details were included in the text, and the manuscript was revised thoroughly. All the modifications performed in the revised manuscript are highlighted. The accompanying document at the bottom of this letter contains a point-to-point reply to the reviewer. We also submitted a revised version of our manuscript on the website.

Thank you in advance for the attention.

Reviewer 1

The manuscript describes current technologies to accelerate the 2 aging process of alcoholic beverages. The manuscript is extensive and clearly written. The table shows the advantages and disadvantages of the new technologies. It also tabulates the main results of recent research, which makes this review scientifically significant. However, a similar review of the literature was recently published, which I would like to include in the manuscript (Soraya Solar, Remedios Castro & Enrique Durán Guerrero (2021): New Accelerating Techniques Applied to the Ageing of Oenological Products, Food Reviews International, DOI: 10.1080/87559129.2021.1934009).

Response: The suggestion was accepted and more information and citations were added to the text. Please check lines 41-53; 116-122; 461-466

I also suggest that more attention be given to describing the effects of new technologies on sensory properties, which may be questionable when using these physical methods. Unfortunately, very often in the manuscripts the results of the sensory analyzes are "hidden", precisely because of the impression of "cooked wine" that can result from the application of these technologies, while only the influence on phenols, color, etc. is indicated (see Jambrak, A.R., Innovative Food Science and Emerging Technologies (2017), http://dx.doi.org/10.1016/j.ifset.2017.04.015). Some of these shortcomings are listed in Table 1 (like metallic flavor), but perhaps this should be emphasized even more.

 Response: The suggestion was accepted and more information and citations were added to the text. Please check lines 263-278 and table 1.

We think we were able to respond adequately to all issues raised by the reviewer and hope you will find our manuscript now acceptable for publication in the Beverages. If there is any other modification to be performed, we will be happy to carry out.

Yours sincerely,

Msc. Roselini Trapp Krüger,

Graduate Program of Food Science and Technology

State University of Ponta Grossa

Reviewer 2 Report

The authors report a literature review on the current methods for the acceleration of alcoholic (mainly wine) beverages.

While the topic of the review is interesting and aims to identify the main lines of research on technology, references included in the review look somehow outdated. If this is not due to insufficient research effort by the authors, it could be the symptom of a low interest in the research world for some technologies. Therefore, the structure of the review should be modified to include a critical view of the technologies adopted, with which the reader can be directed towards the more established methodologies. However, the literature cited appears consistent with the topic addressed and is based on application categorization.

Conclusion
This section appears to be relatively brief and with scant indications. Moreover, the future perspectives paragraph could be included in the conclusions, and the authors could therefore expand this as a unique section through a rewrite of a classification of the methodologies most reported in the literature, focusing on single products or product categories.

Typographical errors
Throughout the paper, some sentences need clarification (as il lines 75-79) or use of different terms (as in line 515). It could be therefore advisable for the final manuscript to be read by a native English speaker, also for typos.

Overall comment
The review shows value but needs a major revision to clarify the points addressed above. It is therefore recommended a critical integration of the above highlighted issues and a critical rewrite of the conclusion section with focus on the most promising technologies.

Author Response

Thank you for your response on our manuscript beverages-1894695.

The constructive criticism of the reviewer was much appreciated and we revised our manuscript accordingly. All suggestions were accepted, more information and details were included in the text, and the manuscript was revised thoroughly. All the modifications performed in the revised manuscript are highlighted. The accompanying document at the bottom of this letter contains a point-to-point reply to the reviewer. We also submitted a revised version of our manuscript on the website.

Thank you in advance for the attention.

Reviewer 2

  • While the topic of the review is interesting and aims to identify the main lines of research on technology, references included in the review look somehow outdated. If this is not due to insufficient research effort by the authors, it could be the symptom of a low interest in the research world for some technologies. Therefore, the structure of the review should be modified to include a critical view of the technologies adopted, with which the reader can be directed towards the more established methodologies. However, the literature cited appears consistent with the topic addressed and is based on application categorization.

Response: The suggestion was accepted and more information and citations most recents were added to the text and tables.

  • Conclusion
    This section appears to be relatively brief and with scant indications. Moreover, the future perspectives paragraph could be included in the conclusions, and the authors could therefore expand this as a unique section through a rewrite of a classification of the methodologies most reported in the literature, focusing on single products or product categories.

Response: The suggestion was accepted and changes were performed accordingly. Please check lines 558-573

  • Typographical errors
    Throughout the paper, some sentences need clarification (as il lines 75-79) or use of different terms (as in line 515). It could be therefore advisable for the final manuscript to be read by a native English speaker, also for typos

Response: The suggestion was accepted and changes were performed accordingly. Please check lines 71-76 and 558-573.

The final manuscript, after revisions, was read by a native English speaker. If a certificate of attestation is required, it can be obtained.

We think we were able to respond adequately to all issues raised by the reviewer and hope you will find our manuscript now acceptable for publication in the Beverages. If there is any other modification to be performed, we will be happy to carry out.

Yours sincerely,

Msc. Roselini Trapp Krüger,

Graduate Program of Food Science and Technology

State University of Ponta Grossa

Reviewer 3 Report

The manuscript is interesting, but some changes should be done to improve the overall quality:

A brief methodology section should be included after Introduction section. In this section, authors should explain the procedure done for the references identification and collection from which scientific databases. It was defined a period of time for the publications analyzed (eg. 2000 to 2022)? If yes, it should be referred.

Pg2. Line 71 – Please replace the sentence “and contact time between wine and Wood” by “and contact time between wine and wood”.

In Table 1, a column with all bibliographic references should be added, line by line (like it was done in tables 2 and 3).

I strongly recommend authors to include the main information of most recent published references in the manuscript, such as:

Note: I am not author or coauthor of these papers. The papers are very valuable and can deeply benefit the quality of the manuscript. 

Sara Canas; Ofélia Anjos; Ilda Caldeira; Tiago A. Fernandes; Nádia Santos; Sílvia Lourenço; Joana Granja-Soares; et al. "Micro-oxygenation level as a key to explain the variation in the colour and chemical composition of wine spirits aged with chestnut wood staves". LWT (2022): 112658-112658. https://doi.org/10.1016/j.lwt.2021.112658.

Sheila Oliveira-Alves; Sílvia Lourenço; Ofélia Anjos; Tiago A. Fernandes; Ilda Caldeira; Sofia Catarino; Sara Canas. "Influence of the Storage in Bottle on the Antioxidant Activities and Related Chemical Characteristics of Wine Spirits Aged with Chestnut Staves and Micro-Oxygenation". Molecules (2021): https://doi.org/10.3390/molecules27010106.

Ilda Caldeira; Cláudia Vitória; Ofélia Anjos; Tiago A. Fernandes; Eugénia Gallardo; Laurent Fargeton; Benjamin Boissier; Sofia Catarino; Sara Canas. "Wine Spirit Ageing with Chestnut Staves under Different Micro-Oxygenation Strategies: Effects on the Volatile Compounds and Sensory Profile". Applied Sciences (2021): https://doi.org/10.3390/app11093991.

J. Granja-Soares; Rita Roque; M.J. Cabrita; Ofélia Anjos; A.P. Belchior; Ilda Caldeira; Sara Canas. "Effect of innovative technology using staves and micro-oxygenation on the odorant and sensory profile of aged wine spirit". Food Chemistry 333 (2020): 127450-127450. https://doi.org/10.1016/j.foodchem.2020.127450.

Canas, Sara; Danalache, Florina; Anjos, Ofélia; Fernandes, Tiago A.; Caldeira, Ilda; Santos, Nádia; Fargeton, Laurent; Boissier, Benjamin; Catarino, Sofia. "Behaviour of Low Molecular Weight Compounds, Iron and Copper of Wine Spirit Aged with Chestnut Staves under Different Levels of Micro-Oxygenation". Molecules 25 22 (2020): 5266-5291. http://dx.doi.org/10.3390/molecules25225266.

Author Response

Thank you for your response on our manuscript beverages-1894695.

The constructive criticism of the reviewer was much appreciated and we revised our manuscript accordingly. All suggestions were accepted, more information and details were included in the text, and the manuscript was revised thoroughly. All the modifications performed in the revised manuscript are highlighted. The accompanying document at the bottom of this letter contains a point-to-point reply to the reviewer. We also submitted a revised version of our manuscript on the website.

Thank you in advance for the attention.

Reviewer 3

  • A brief methodology section should be included after Introduction section. In this section, authors should explain the procedure done for the references identification and collection from which scientific databases. It was defined a period of time for the publications analyzed (eg. 2000 to 2022)? If yes, it should be referred.

Response: The suggestion was accepted and changes were performed accordingly. Please check lines 89-97.

  • Line 71 – Please replace the sentence “and contact time between wine and Wood” by “and contact time between wine and wood”.

Response: The suggestion was accepted and changes were performed accordingly.

  • In Table 1, a column with all bibliographic references should be added, line by line (like it was done in tables 2 and 3).

Response: The suggestion was accepted and changes were performed accordingly.

  • I strongly recommend authors to include the main information of most recent published references in the manuscript, such as:

Response: The suggestion was accepted and more information and citations were added to the text.

Please check lines 302-312; 313-319 and table 3.

We think we were able to respond adequately to all issues raised by the reviewer and hope you will find our manuscript now acceptable for publication in the Beverages. If there is any other modification to be performed, we will be happy to carry out.

Yours sincerely,

Msc. Roselini Trapp Krüger,

Graduate Program of Food Science and Technology

State University of Ponta Grossa

Round 2

Reviewer 2 Report

The authors addressed all relevant observations made by this reviewer and paper overall quality improved. Under these circumstances, there is no more objection to publication by this reviewer.

Reviewer 3 Report

Dear authors,

I really appreciate the effort in improvements that have been indicated. The manuscript is now more complete and updated.

Thank you.